# Community Participation in the Importance of Living Heritage Conservation and Its Relationships with the Community-Based Education Model towards Creating a Sustainable Community in Melaka UNESCO World Heritage Site

Noor Azramalina Abdul Aziz [1], Noor Fazamimah Mohd Ariffin [1,*], Nor Atiah Ismail [1] and Anuar Alias [2]

[1] Department of Landscape Architecture, Faculty of Design and Architecture, Universiti Putra Malaysia, Serdang 43400, Malaysia

[2] Department of Estate Management, Faculty of Built Environment, Universiti Malaya, Jalan Universiti, Kuala Lumpur 50603, Malaysia

\* Correspondence: fazamimah@upm.edu.my; Tel.: +60-123905569

**Abstract:** Living heritage runs the risk of being lost forever, or frozen as a practice of the past, if not promoted in the community. The preservation of this history, its transmission to following generations, and its ability to transform and adapt to any circumstance, are all made possible by strengthening living heritage. Investigating the function of living heritage in advancing education for sustainable development has been deemed a crucial goal by quality education as Sustainable Development Goal number 4 (SDG 4). The aim of this article is to gather information on living heritage conservation toward creating a sustainability community by using the community-based education model on the communities' attitudes, cultural knowledge, and awareness of the importance of living heritage, and their participation level towards living heritage conservation in Melaka UNESCO World Heritage Site. This study uses the quantitative method of online questionnaire survey technique to collect data. There are 392 respondents from the multicultural community of Melaka World Heritage Site, who randomly responded. Based on the mean comparison in gender, age level, and race, there is a positive significant relationship between the importance of living heritage and the local community's participation level. The increasing of the participation level to ACTIVE would lead to a higher altitude, cultural knowledge, and awareness of the importance of living heritage in the local community.

**Keywords:** intangible cultural heritage; public awareness; quality education; sustainable development

## 1. Introduction

Preservation is defined as an effort to maintain cultural materials in both tangible and intangible forms, including oral tradition, music, and cultural activities. Tangible cultural heritage includes elements such as buildings, landscapes, structures, locations, and communities [1,2].

The most effective strategy for promoting cultural diversity is intangible cultural heritage (ICH), which demonstrates the diversity of living human heritage. ICH's "self-identification as an imperative part of its creator's and carriers' cultural identity" determines its main "constitutive components". The ICH fixed recreation in feedback to the historical and social transformation of the communities and groups in question by connecting with their cultural identity, authenticity, and unbreakable connection to human rights [3]. The depth of meaning, attachment, and variety of place experiences are all impacted by place identity, which is linked to how people define and perceive their surroundings. The traditional settings of the city center are being altered by new developments, as shown by this evidence [4], which also shows how place definitions and extensions are being changed.

Influential 20th-century thinker, R. Williams, stated in 1960 that culture could not be compressed into tangible elements since it lives and changes constantly. He was able to capture the historical legacy of the fundamental components defining the ongoing evolution and development of human communities. Cultural heritage as people's way of life encompasses all supplementary elements that a given community views as mandatory parts of its inherent identity, as well as its uniqueness and distinctiveness in relation to all other human groups, demonstrating the very essence of its distinctive trait.

The United Nations' Organizations Agenda for Sustainable Development (Agenda 2030) supports the aforementioned claim and lists "safeguarding the world's cultural and natural legacy" as one of its goals in item 11.4 of the document [5]. The Sustainable Development Goals (SDG) and the fundamental foundations of the Roadmap on Education for Sustainable Development 2030 are both emphasized once more [6,7].

Amid a time of increasing complexity, change, and a lack of predictability, to transform how information and learning may benefit humanity, the United Nations Educational, Scientific, and Cultural Organization (UNESCO) launched the Future of Education initiative in September 2019. UNESCO focuses on challenges that will have an impact on the well-being and education of future generations, such as climate change, growing imbalances, artificial intelligence, educational outcomes, and opportunities. UNESCO is currently inviting people from all across the world to share their visions for the future of knowledge.

The Future Education Initiative's goal is to comprehend how education might shape humanity's and the planet's futures in 2050, and beyond [8]. The initiative is generating a global debate on how knowledge, teaching, and learning must change to solve today's and tomorrow's issues, involving youth, educators, civil society, governments, businesses, and other stakeholders. The idea of the future is used in this effort to highlight the great range of ways people all around the world know, and therefore are.

In the Fourth Industrial Revolution, artificial intelligence, and in a robotics-driven modern world, we must promote flexibility and adaptation. As a result, education needs to be entirely rethought. Instead of memorizing facts and figures, people should "learn how to study" and find solutions to problems [9]. They should be inspired to investigate independent learning as well. Changes are needed on every level. To prepare individuals for the future, we must develop an education system that is both backward-looking and forward-looking.

Previous researchers worked together to develop specific guidelines [10] and mapped learning and teaching practices and policies on living heritage, within the context of education for sustainable development [11–14], in order to organize learning and teaching with a living heritage for a sustainable future.

Participation in local communities is one of the crucial aspects for the management of WHS. The Melaka city area has been a UNESCO WHS for 13 years, so what is the contribution of community and government doing to safeguarding their intangible cultural heritage? The importance of community participation throughout decision-making, implementation, and enforcement, has been widely acknowledged [15,16]. However, Ong (2017) research recognized that this aspect is the most neglected by the authorities concerned in Melaka WHS. This situation affects the attitude of the local community to conserve their heritage. One of the critical success factors for sustainable conservation is the awareness and appreciation of the heritage value of the resources by stakeholders, particularly the local community. An informed society or community will make wise decisions about protecting and preserving resources that define the very essence of their culture and society [17].

The success of heritage site conservation depends on two factors, which are the stakeholders' awareness, participation, and appreciation of heritage values and their economic potential [17,18], and the public education programs designed for various stakeholders [17].

Educational opportunities and programs are scarce in Melaka WHS regarding the participation process in conserving the living heritage site in Melaka WHS [19]. Rahimah (2017) states that they have had to deal with difficulties, including the modernization and development taking place in Melaka State, which is viewed as increasing the vulnerability

of their existence or survival, as well as concerns about authenticity. Additioanlly, there are difficulties concerning the participation and involvement of members of the respective community in cultural heritage conservation. The younger generation are slowly losing their ethnic heritage and identity under the onslaught of modernization and globalization in favor of adopting more mod values [20]. The communities would lose their ethnic identity and cultural values because of these external influences, as the modern way of life dominates traditional knowledge.

This article advocates studying the community's attitudes, cultural knowledge, and awareness of living heritage site conservation in Melaka UNESCO World Heritage Site (WHS), as the third step in the community-based education (CBE) model of information gathering. By enhancing, protecting, and passing down the living heritage that ensures its continuity to future generations, this study aimed to create a sustainable community.

## 2. Literature Review

### 2.1. Intangible Cultural Heritage

The 2003 Convention on the Safeguarding of Intangible Cultural Heritage [21] lists five broad "domains" in which intangible cultural heritage (ICH) emerges. These five criteria will be used to assess a participant's competency and comprehension in light of daily standards, from both their own and another culture. Intangible cultural heritage is described in Article 2 of the UNESCO Convention as representations, expressions, knowledge, skills, and the accompanying tools, artifacts, and cultural sites. The communities viewed themselves as representatives of their cultural heritage. If it is not preserved in the community, intangible cultural heritage faces the risk of becoming extinct or a legacy of the past. Therefore, it is essential to safeguard intangible cultural heritage, as it keeps it alive, transmits it to future generations, and offers it the adaptability to face any situation. Safeguarding ICH for future development is essential for the economy, society, and environment. It also fosters peace and security. The greatest technique to assist local populations and the community is to transmit ICH traditions, or living heritage, through excellent instruction.

The 2003 UNESCO Convention for the Protection of Intangible Cultural Heritage or Living Heritage proposes five broad domains, namely [21]:

(1) Oral traditions and expressions encompass a wide variety of speech forms, including proverbs, riddles, fables, nursery rhymes, stories, myths, epic songs and poems, charms, prayers, chants, songs, dramatic performances, and more. Through it, information, cultural and social values, and collective memory, are all transmitted. It is also essential for preserving tradition.

(2) The performing arts include chanting, pantomime, vocal and instrumental music, dancing, and theatre. It contains several cultural manifestations that showcase human creativity.

(3) Social practices, rituals, and festivals are commonplace routines that shape community and group life and are essential to all participants. It is crucial because it represents the group or society's identity and is strongly tied to significant occasions, whether they take place in public or private settings.

(4) Knowledge and practices about nature and the universe comprise the community's developed knowledge, abilities, behaviors, and representations that interact with the environment. Language, oral traditions, sentimental ties to a place, memories, spirituality, and worldview are examples of how this method of understanding the universe is expressed. It also has a significant impact on attitudes and beliefs, as well as numerous social traditions and cultural activities.

(5) Traditional craftsmanship is expressed most succinctly in intangible cultural heritage. It emphasizes the skills and knowledge needed for carpentry rather than the craft of the end product. It must therefore inspire artisans to continue producing their work and passing on their skills to others, particularly those within their communities.

### 2.2. Community-Based Education (CBE) Model

CBE's goal is to empower adults and youth in the community by promoting involvement and education, as well as by identifying and resolving [22] conservation issues relevant to the community's living legacy, in the context of local social and economic elements. In other words, community-based behavior is inspired by education. One of the best results achieved by CBE is the building of collaboration ability to satisfy their common goals in community development plans [23]. Instead of only the definition of "education based in the community", activities are concentrated on four key characteristics: community-based, collaborative, information-based, and action-oriented.

The effectiveness of the initiative is influenced by the community's background, including member education, participation, place-based, youth and community development in a diversity of productive activities [24]. Five phases are suggested, based on the model in Figure 1, for capacity building using community-based education concepts. The three-step informed group activities for sustainable CBE management and the planning process for LH conservation, which is where the detailed process used in this research study is located in this paper. The stakeholders must complete the tasks to assess strengths, assess needs, gather data, and plan actions before effective education may be achieved. As a result, this study will primarily focus on the information-gathering phase.

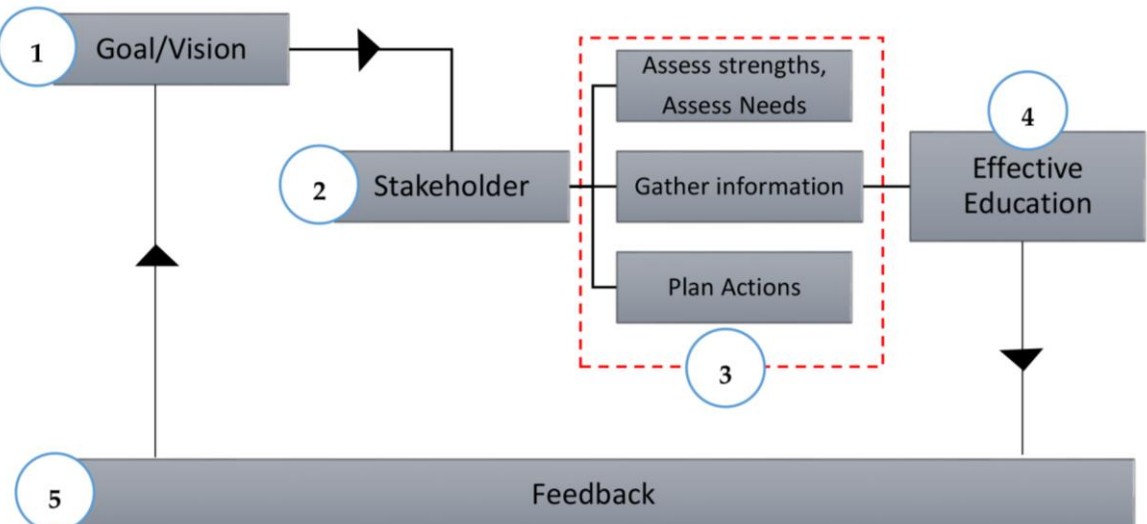

**Figure 1.** Building capacity: applying the principles of community-based education for living heritage conservation site, readapted with permission from Ref. [25].

Informed Group Activities

There are three divisions in informed group activities, which are: (1) assessing the community's strengths and needs; (2) gathering information; and (3) planning actions. This section was essential to identify the local communities' perception of their ICH value and to create effective education from analysis results. Therefore, this paper will be focused only on the gathering information part on the communities' attitudes, cultural knowledge, and awareness of the importance of living heritage, and their participation level towards living heritage conservation in Melaka UNESCO World Heritage Site.

i.        Assess the Community's Strengths and Needs

In any community capacity-building approach, community strengths and needs are essential factors if stimulated by a goal [26]. This study investigates the current status of the strength of the community in living heritage conservation areas on heritage resources such as history, visual and performing arts, heritage buildings, landscapes, and unique lifestyles, values, traditions, and events.

Understanding the needs of the entire community is the foundation of any approach to building community capacity. Individual needs will differ from person to person in terms of complexity and strength. Certain requirements may be more broadly shared within the community when it comes to gaining access to services and/or resources [27]. A community's changing political, social, and economic conditions will affect collective demands to varying degrees [28–30]. We can decide how the community can assist a particular person to handle their challenges more successfully by concentrating on that person's needs [31]. Authorities can better understand the community by interacting with community members directly about their needs, in order to enhance decision-making for sustainable development.

ii.    Gather Information on Community's Attitude, Cultural Knowledge, and Awareness toward Living Heritage Conservation

Understanding attitudes, cultural knowledge, and awareness of LH conservation are necessary to determine the amount of knowledge and awareness of the local community's own ICH in everyday practice. When evaluating stakeholders, the attitudes, cultural awareness, and a general understanding of the living heritage area, are essential factors in determining the sustainability of a heritage site [32–34]. This study will first ascertain the community's attitude toward, the amount of knowledge about, and an awareness of ICH. The wealth of knowledge and skills passed down from one generation to the next was essential for ICH conservation. The importance of knowledge's social and economic value is significant to both minority groups and groups in the majority society. Processes, words, knowledge, skills, connected artifacts, and cultural spaces are examples of intangible cultural heritage that individuals recognize as a part of their own. Being passed down through the generations and continually renewed, the spread gives humans a sense of identity and continuity. Examples of factors that have a favorable impact on the economy and social development include growth and development rates, foreign exchange outflow volumes, infrastructure development, innovative management practices, and training experience. A valuable economic source is the preservation of intangible cultural assets [35]. Therefore, wherever possible, the society, people, and, as necessary, the specific persons who represent such a heritage, must be included in conservation efforts.

Table 1 shows that intangible cultural heritage contributes to inclusive social development [16,36], environmental sustainability [16,37], inclusive economic development [16,35], and peace and security [16,38,39].

Second, based on the level of participation, the local community's attitude, knowledge, and awareness of LH conservation will be used to identify WHS conservation initiatives and cultural heritage education in this study. It has long been understood how crucial community participation is in the decision-making, implementation, and enforcement processes [39]. Numerous forms of community participation, from manipulative collaboration to citizen power, are discussed in the literature [40–44]. As a pioneer in this field, Arnstein (1969) proposed an eight-tier ladder of community participation divided into three categories: manipulative participation, citizen tokenism, and citizen power. Similar to this, Pretty (1995) created a typology of community involvement, which included manipulative participation, passive participation, and self-mobilization [42,45]. Tosun (1999, 2006) described three types of community participation in tourism development: coercive participation, induced participation, and spontaneous participation [45], combining the typologies of Arnstein (1969) and Pretty (1995).

Coerced residents are only marginally involved in development activities and have little control over decision-making or oversight [44,46]. Instead, governmental organizations and the private sector are in charge of monitoring the development of the tourism business [44,46]. Only by informing the community of planned projects and how those developments can benefit them can local governments engage the community [47]. According to Zhang et al. (2013), those in positions of power just need to tell the community about initiatives in order to fulfill their legal obligations and placate locals. In turn, this reduces opposition to the planned change. The residents' participation and opinions, however,

are not appreciated, and they have little practical power to influence the direction of the development [46].

**Table 1.** Intangible cultural heritage contributes to social, environmental, economic, and peace and security.

| Contribution Category | Finding | Author | | | | | |
|---|---|---|---|---|---|---|---|
| | | UNESCO, 2015 [16] | Petronela, 2016 [35] | S.-K. Tan et al., 2018 [36] | Ounanian, K. et al., 2021 [37] | Agarwal, S., 2018 [38] | UNESCO 2014 [39] |
| Inclusive Social Development | Intangible cultural heritage is vital to achieving food security. | ✓ | | | | | |
| | Traditional health practices can contribute to the well-being and Inclusive quality of health care for all. | ✓ | | | | | |
| | Traditional practices concerning water management can contribute to equitable access to clean water and sustainable water use. | ✓ | | | | | |
| | Intangible cultural heritage provides living examples of educational content and method. | ✓ | | | | | |
| | Intangible cultural heritage can help strengthen social cohesion and inclusion. | ✓ | | | | | |
| | Intangible cultural heritage is decisive in creating and transmitting gender roles and identities and, therefore, critical for gender equality. | ✓ | | | | | |
| | Intangible cultural heritage as a place attachment, sense of place, and place identity. | | | ✓ | | | |
| | Sense of loss when a lack of transmission of intangible cultural heritage knowledge and skills. | | | ✓ | | | |
| Environmental Sustainability | Intangible cultural heritage can help protect biodiversity. | ✓ | | | ✓ | | |
| | Intangible cultural heritage can contribute to environmental sustainability. | ✓ | | | ✓ | | |
| | Local knowledge and practices concerning nature can contribute to the research on environmental sustainability. | ✓ | | | ✓ | | |
| | Knowledge and coping strategies often provide a foundation for community-based resilience to natural disasters and climate change. | ✓ | | | ✓ | | |
| Inclusive Economic Development | Intangible cultural heritage is often essential to sustaining the livelihoods of groups and communities. | ✓ | ✓ | | | | |
| | Intangible cultural heritage can generate revenue and decent work for many people and individuals, including poor and vulnerable ones. | ✓ | ✓ | | | | |
| | Intangible cultural heritage, as a living heritage, can be a significant source of innovation for development. | ✓ | ✓ | | | | |
| | Communities can also benefit from tourism activities related to intangible cultural heritage. | ✓ | ✓ | | | | |
| Peace & Security | Many intangible cultural heritage practices promote peace at their very core. | ✓ | | | | | ✓ |
| | Intangible cultural heritage can help to prevent or resolve disputes. | ✓ | | | | | |
| | Intangible cultural heritage can contribute to restoring peace and security. | ✓ | | | | ✓ | ✓ |
| | Protecting intangible cultural heritage is also a means to lasting peace and security. | ✓ | | | ✓ | | |
| | Intangible cultural heritage in conflict-related emergencies. | | | | | | ✓ |

The second form of community participation, according to Tosun's (2006) typology, is induced community engagement, which is related to citizen tokenism in Arnstein's (1969) model and passive participation in Petty's (1995) typology. Despite the fact that they have a voice in the tourism development process and that decision-makers do pay attention to their ideas, residents do not actually have any impact or authority over the decision-making

process in induced community participation [44,46]. The decision-makers have the last word on whether to accept or reject suggestions made by residents during the planning and development process [45]. This type of community participation, also known as a public hearing or community consultation [48], usually occurs later in the planning process, after the majority of the concerns and options have been considered.

The various types of participation processes are shown in Table 2. Tosun (2006) describes the highest level of community participation as a spontaneous participation, Arnstein (1969) refers to it as citizen power, and Pretty (1995) refers to it as self-mobilization. Residents have the ability to make decisions and manage the development process through spontaneous participation. Spontaneous participation has the potential to increase resident trust, ownership, and social capital in contrast to the other two forms of conventional participation, which do not constitute effective participation and lead to conflicts [45,49]. All resident and stakeholder groups are actively involved throughout the entire participatory planning process due to spontaneous participation, which starts in the early stages of the planning process [44,48].

**Table 2.** The different types of participation processes.

| Type<br>Components | Coercive Participation | Induced Participation | Spontaneous Participation |
|---|---|---|---|
| **Level of Participation** | Low level/Passive | Middle level/Responsive | High level/Active |
| **Involvement** | Negligible involvement (limited) | Passive involvement | Active involvement |
| **Action** | No actual power to make the decision and to control the development process. | No actual power to make decisions and control the development process. | Have the power to make decisions and control the development process. |
| **Time involvement** | Just get the information. | Usually, happen after development. | Early planning stage. |
| **Input** | Government, authorities, and the private sector exert their control. | Public hearing or community consultation. | Residents can generate trust, ownership, and social capital. |

iii.   Plan Action based on Community Perceptive

From the finding of assessing the community's strengths and needs, and gathering information on the community's attitude, cultural knowledge, and awareness toward living heritage conservation, the plan of action based on the community perceptive is the next step that will be investigated later, based on Cultural Heritage Education Programs (CHEP) in four case studies: Penang (Malaysia) [50], Singapore [51], the Philippines [52], and Europe [53], to identify the elements of learning content, learning preferences, and teaching-learning technique in this study. In addition, the comparison of the best practices of community-based education for living heritage site conservation in these four case studies was made by Aziz et al. [54].

*2.3. CBE for LH toward Sustainable Community*

In a sustainable community, multiple human needs are taken into account and satisfied, not just one at the expense of the others [55]. It is a setting where people from all backgrounds and perspectives can feel comfortable and welcomed, where all groups can take part in decision-making, and where prosperity is shared. A sustainable community balances the requirements of the present with the conservation of sufficient economic, social, and environmental resources for future generations [56,57]. There are eight component keywords to create a sustainable community, which is a community [58,59] that is: (1) well run; (2) active, inclusive, and safe; (3) sensitive to the environment; (4) thriving; (5) fair for all; (6) well connected; (7) well served; and (8) well designed and built (shown in Figure 2). Through the CBE model for LH, showing a well-run community is a first and second step to involving local people in all community-to-community decision-making processes,

forming a vision, and overall enjoying civic values, responsibility, and pride for achieving the goal of survival of the living heritage. A significant part of CBE for LH is establishing a community's vision and goals since it forms the basis for a strategy consultation that provides guidance on how and when the strategy might be used, either independently or in collaboration with other strategies [60]. As a result of this formulation, the vision and goals of CBE for LH will be formed as a guide for providing a top-notch education.

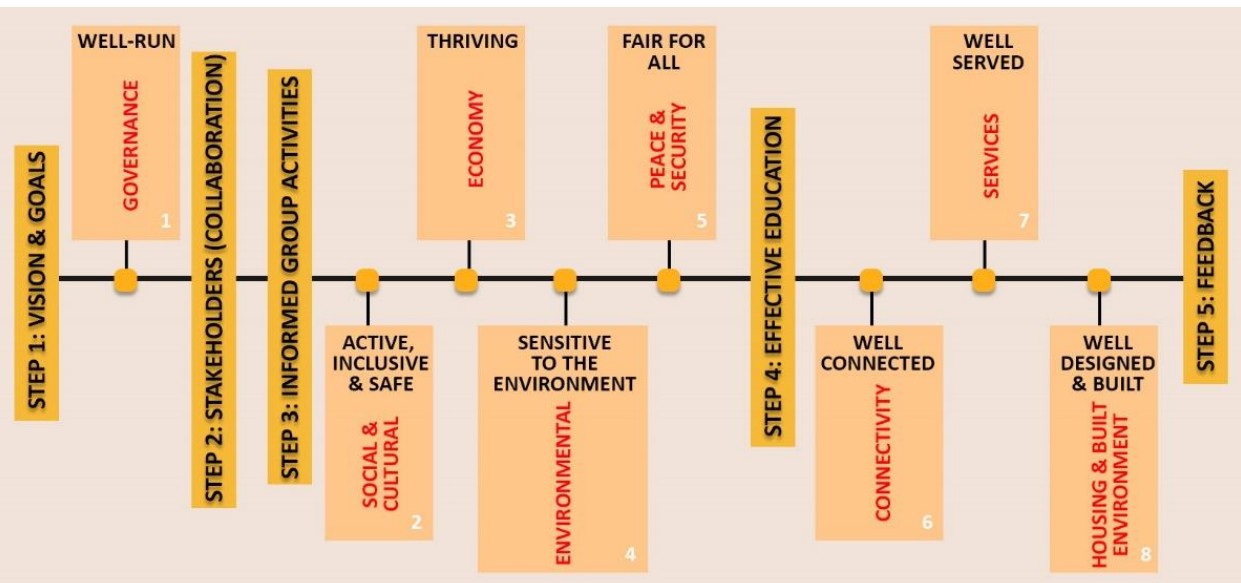

**Figure 2.** Building capacity: applying the principles of community-based education for living heritage toward sustainable community.

The third step in the CBE model for LH is information-gathering activities to identify the level of community strength, community needs, and community basic knowledge about cultural heritage. Through this step, the data of the results of each level of strength, need, and knowledge, and will take action plans based on the perspective of the community to create an active, inclusive, and safe community in social and cultural, thriving, sensitive to the environment, and fair for all. An active, inclusive, and safe community is a true sense of community where neighbors look out for each other, welcome to join events, and where there is healthy respect between cultures, and all are treated equally. In general, the community has the opportunity to earn money and achieve a good quality of life through the knowledge and skills of the living heritage provided by encouraging the community to open local businesses, create jobs for others, and spend and invest locally in a thriving community. Environmentally sensitive communities are communities that practice healthy lifestyles by actively trying to minimize climate change by encouraging recycling, water conservation, and by maintaining a cleaner, safer, and greener neighborhood. A fair community for all is where every individual of all ages, races, gender, and disabilities, is taken into account and given equal access to jobs, services, and education in the community.

The fourth step in the CBE model for LH is the formation of effective education where the knowledge and skills of living heritage provided and taught will create a community that is well-connected, well-served, and well-designed and built. A well-connected community promotes safe walking on heritage trails while connecting neighborhoods between communities and has communications connecting people to jobs, health, and other services. A well-served community is a community that provides good services in terms of easy access to fresh food and raw materials, high-quality and accessible family services, including healthy options for the community that is available and affordable, and volunteer and private services. Finally, a well-designed and well-built community has a real sense of place, positive purpose, and local character where heritage buildings are not

only attractive but also safe. It also has local activities that are still maintained and valuable and provide plenty of green and open space for people to spend time relaxing and playing. The fifth step in the CBE for LH model is the feedback that influences the vision and goals for quality education for living heritage. It ensures that CBE for LH runs smoothly and makes improvements if necessary.

## 3. Materials and Methods

This study uses the quantitative method of questionnaire survey technique to collect data on the communities' attitudes, cultural knowledge, and awareness of the importance of living heritage and their participation level towards living heritage conservation in Melaka UNESCO World Heritage Site. There were 392 respondents from the multicultural community of Melaka World Heritage Site, who randomly responded in July 2020.

### 3.1. Sample and Sampling Method

The survey tool for this study is an online questionnaire created with Google Forms. The questions were chosen because it related to the category of people's opinions, attitudes, and behavior [61]. Additionally, a questionnaire can minimize the interviewer's influence on the participants' responses [61]. According to the Department of Statistics Malaysia [62], the local community of Melaka WHS has respondents ranging in age from 15 to 64 years old who are of working age and are able to understand and express ideas.

### 3.2. Survey Instrument

The questionnaire was divided up into several sections, with questions about: (i) demographic data, (ii) the significance of LH, and (iii) the level of participation. To help answer the research questions about the level of community members' views, cultural knowledge, and awareness, the questionnaire included closed-ended questions. While the participation level comprises six (6) items across three (3) stage levels, the importance of LH has ten (10) items across four (4) factors of contribution. The 5-point Likert scale was used for the entirety of the response options in this study. The Likert Scale is an illustration of a composite assessment used to improve measurement standards in social research [63]. The scale in this study includes standard response categories such as "strongly disagree", "disagree", "partially agree", "agree", and "strongly agree", in order to determine the relative weight of each item.

### 3.3. Data Analysis

The data was transformed to digital form using SPSS 22.0. Both "user missing" (data that was absent during analysis and modification) and "missing system" types of missing data exist (utterly absent from the data as the respondent fails to answer it). In this work, there were no user-missing data. To determine if the missing data in the system was random, the author applied missing value analysis. It found out that it was. Similar to this, the patterns of the missing data had been investigated; however, no systematic pattern had been found, and the missing data were random [64].

To confirm that the data was internally consistent, a reliability test was carried out. In order to achieve the research objective, descriptive analysis is used to determine the demographic differences among respondents, the mean value of LH, and the level of participation. According to several studies, mean values are the best method for analyzing data from Likert scales as far as the validity of the analyses is concerned [65,66].

To check for a strong relationship and significance between the data, a correlation was conducted. Reliability is the measure of the internal consistency of the constructs in this study. A construct is reliable if the Alpha ($\alpha$) value is greater than 0.70 [67]. Construct reliability was assessed using Cronbach's Alpha. The results revealed that the participation level scale with six items ($\alpha = 0.882$) and the importance of the LH scale with ten items ($\alpha = 0.944$). Reliability results are shown in Table 3.

**Table 3.** Reliability statistics of the participation level and the importance of LH.

| Constructs | No. of Items | Alpha ($\alpha$) |
|---|---|---|
| Participation level | 6 | 0.882 |
| Importance of LH | 10 | 0.944 |

## 4. Results and Discussions

The continued use of heritage by the community it is linked with for the original purpose for which it was created is what is known as "Living Heritage" (LH). It has a unique bond to a community. As change is accepted as a component of the living nature of the heritage place, it is exposed to a continuous process of evolution as a result [68]. Melaka was chosen as a case study for this study. Melaka State, which is situated on the west coast of the central Peninsular of Malaysia, is bordered to the west by the Straits of Malacca, to the north by the State of Negeri Sembilan, and to the south by the State of Johor (shown in Figure 3).

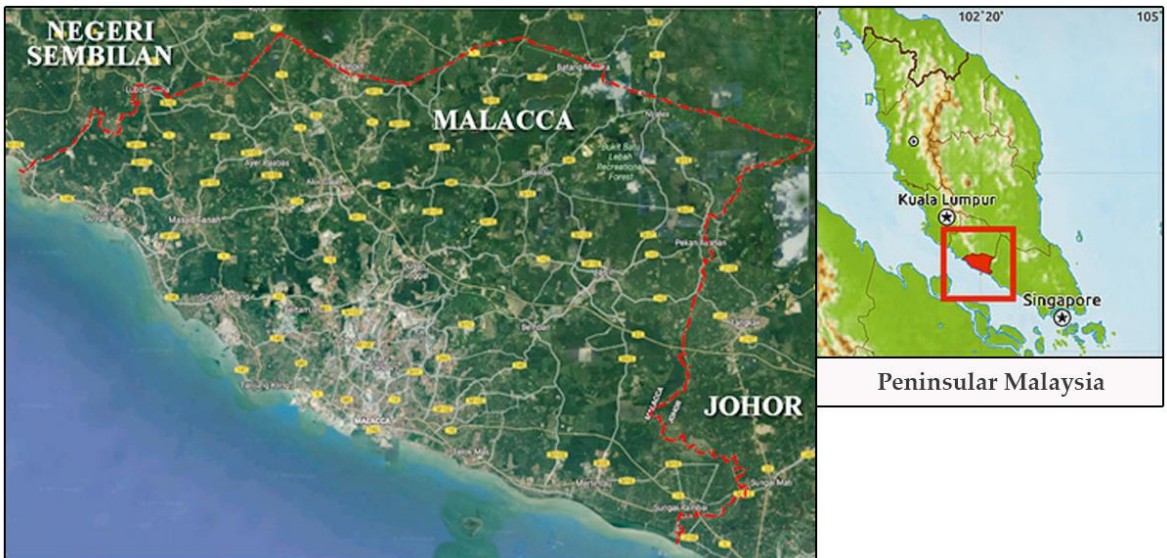

**Figure 3.** Location of the study.

The city of Melaka and George Town in Penang was designated as UNESCO World Legacy Sites on July 7, 2008, in recognition of their distinctive multicultural living heritage that dates back to the trade routes and the outstanding universal values (OUV). The historic cities of the Straits Settlements are comprised of them all. The following is how UNESCO described Melaka's outstanding universal values in the inscription as a WHS [68]:

- Remarkable displays of multicultural trade towns formed by the commercial exchanges of Malay, Chinese, Indian, and European cultures, as well as the influences of architecture, urban form, technology, and monumental art;
- A tangible and intangible manifestation of the colonial influences and the multicultural heritage of Asia and Europe, exemplified by the diversity of religious structures of various faiths, ethnic communities, numerous languages, worship and religious festivals, dances, costumes, art, music, food, and daily life;
- A mixture of elements that have created an unmatched architecture, culture, and urban environment in East and South Asia. Primarily the unique variety of townhouses and shophouses, each in a different stage of development.

The Melaka city region was needed to safeguard its distinctive tangible and intangible cultural characteristics due to its designation as a World Heritage Site. As a result, altering or destroying the look of its historic structures is prohibited. Melaka also needs to raise awareness of the WHS, particularly among its communities, foster a sense of ownership, and build support by highlighting its importance. The estimated population distribution

by ethnic group in Melaka is shown using data from Table 4 [69]. Three hundred and eighty-four (384) samples are needed to represent one million populations with a 95% confidence level and a 5% error margin. A total of three hundred and ninety-two (392) respondents were chosen at random to participate in the data collection.

**Table 4.** Melaka—estimated population by ethnic group, readapted data with permission from Ref. [69].

| Ethnic Group | 2020 | | Sample Size | Actual |
| --- | --- | --- | --- | --- |
| | Percentage (%) | Population | Data Collection | Data Collection |
| Malay | 71.7 | 715,872.9 | 275 | 268 |
| Chinese | 22.1 | 220,652.6 | 85 | 50 |
| Indian | 5.6 | 55,912.0 | 22 | 13 |
| Others | 0.6 | 5990.6 | 2 | 61 |
| Total | 100 | 998,428 | 384 | 392 |

The results for this paper focused on gathering information, part of which is respondents' attitudes, cultural knowledge, and awareness of the importance of LH and the participation level. Data collection for this study used an online google form questionnaire and was randomly responded from the Melaka local community.

There are three hundred and ninety-two respondents. In the demographic data for gender, there are 193 male (49.2%) and 199 female (50.8%) respondents; seven respondents are different in gender. There are five groups of middle age level which is: (1) 15–24, (2) 25–34, (3) 35–44, (4) 45–54, and (5) 55–64. The higher range level of age was 25–34, with 98 respondents (25%). Malay was the higher number of respondents in the race categories with 268 respondents (68.4%), following the estimated Melaka population by ethnic group [67]. Other races included Portuguese, Nyonya baba, and mixed ethnic. These demographic data results (shown in Table 5) are used in the crosstab of the comparison mean in the importance of LH and the participation level.

**Table 5.** Respondents' demographic data in gender, age level, and race (*n* = 392).

| Demographic | Variable | Frequency | Percentages % |
| --- | --- | --- | --- |
| Gender | Male | 193 | 49.2 |
| | Female | 199 | 50.8 |
| Age Level | 15–24 | 79 | 20.2 |
| | 25–34 | 98 | 25.0 |
| | 35–44 | 85 | 21.7 |
| | 45–54 | 94 | 24.0 |
| | 55–64 | 36 | 9.2 |
| Race | Malay | 268 | 68.4 |
| | Chinese | 50 | 12.8 |
| | Indian | 13 | 3.3 |
| | Others | 61 | 15.6 |

*4.1. Respondents' Attitude, Cultural Knowledge, and Awareness of the Importance of Living Heritage*

There are 10 variables of the importance of LH (shown in Table 6), which are four variables in social contribution (SC1, SC2, SC3 and SC4), two variables in economic contribution (EC1 and EC2), two variables in environmental contribution (EN1 and EN2), and two variables in peace and security contribution (PS1 and PS2). Nine out of 10 variables mean they are in the high level (4: agree), just one variable was in the moderate level (3: partially agree) in the environmental contribution. The EN1: "Knowledge and practice of cultural heritage accumulated over time to make sustainable use of natural resources and minimize the impact of climate change", with a 3.97 mean (red highlighted). This knowledge and practice of cultural heritage accumulated over time to make sustainable use of natural resources and minimize the impact of climate change, needs to be highlighted to

provide information in CBE for LH to create an adequate education. Still, many respondents are unaware that this important variable can change our life if we practice it daily as our routine. The highest mean in social contribution was SC2: the loss of cultural heritage caused losses to the community in Melaka, with a 4.36 mean (blue highlighted). Most of the respondents were aware of the importance of this variable in the community.

**Table 6.** The overall means of attitude, cultural knowledge, and awareness of the importance of LH are based on respondents' perspectives (*n* = 392).

| Importance of LH | Code | Variable | Mean | SD |
|---|---|---|---|---|
| **Social: SC** | SC1 | The cultural heritage in Melaka as my image, identity, and pride. | 4.32 | 0.86 |
| | SC2 | The loss of cultural heritage caused losses to the community in Melaka. | 4.36 | 0.84 |
| | SC3 | I am responsible for practicing my cultural heritage for its continuity in the future. | 4.14 | 0.90 |
| | SC4 | The continuity of heritage culture terminates when there is a lack of transmission. | 4.20 | 0.82 |
| **Economic: EC** | EC1 | Knowledge, skills, and cultural heritage practices contribute to economic improvement and living standards. | 4.14 | 0.85 |
| | EC2 | The originality of the culture is lost, and natural resources are destroyed when there is a lack of awareness in the new development management. | 4.22 | 0.83 |
| **Environmental: EN** | EN1 | Knowledge and practice of cultural heritage accumulated overtime to make sustainable use of natural resources and minimize the impact of climate change. | 3.97 | 0.92 |
| | EN2 | The cultural heritage in Melaka contributes to the continuity between the past, present, and Future in the environment setting. | 4.33 | 0.84 |
| **Peace & Security: PS** | PS1 | Appreciation and understanding of cultural differences between communities create harmony in daily life. | 4.27 | 0.81 |
| | PS2 | An unpeaceful environment occurs when there is a lack of understanding of cultural differences in the community. | 4.18 | 0.83 |

The Comparison Means of Attitude, Cultural Knowledge, and Awareness in Gender, Age Level, and Race of the Importance of Living Heritage

In this subsection, the comparison means of attitude, cultural knowledge, and awareness in gender, age level, and race of the importance of LH are based on respondents' perspectives on social, economic, environmental, and peace and security contributions, are shown in Table 7. Most of the mean of the importance of LH is high level (4: agree), just some variables in moderate level (3: partially agree).

In the gender group, males have a moderate level (3: partially agree) of knowledge awareness of the importance of LH in environmental contribution (EN1) compared to females, that have a high level (4: agree). Therefore, more focus on the male gender must be highlighted to provide information in CBE for LH to create an effective education.

The same variable on the importance of LH in environmental contribution (EN1) has a comparison in the age level group. The age level groups more senior in 45–55 and 55–64 have a high level (4: agree) meanwhile a middle age level group of 15–24, 25–34, and 35–44 have a moderate level (3: partially agree). Recommendation to create an effective education

in CBE for LH, more focus on a middle-age level group of 15–24, 25–34, and 35–44 must be highlighted and taken into consideration.

**Table 7.** The comparison means of attitude, cultural knowledge, and awareness in gender, age level, and race of the importance of LH is based on respondents' perspectives on social, economic, environmental, and peace and security contributions.

| | N | SC1 | SC2 | SC3 | SC4 | SC | EC1 | EC2 | EC | EN1 | EN2 | EN | PS1 | PS2 | PS |
|---|---|---|---|---|---|---|---|---|---|---|---|---|---|---|---|
| **Overall** | 392 | 4.32 | 4.36 | 4.14 | 4.20 | 4.25 | 4.14 | 4.22 | 4.18 | 3.97 | 4.33 | 4.15 | 4.27 | 4.18 | 4.23 |
| **Gender** | | | | | | | | | | | | | | | |
| Male | 193 | 4.31 | 4.33 | 4.09 | 4.21 | 4.24 | 4.10 | 4.24 | 4.17 | 3.91 | 4.31 | 4.11 | 4.26 | 4.13 | 4.20 |
| Female | 199 | 4.33 | 4.39 | 4.18 | 4.19 | 4.27 | 4.18 | 4.21 | 4.19 | 4.04 | 4.36 | 4.20 | 4.29 | 4.22 | 4.25 |
| **Age Level** | | | | | | | | | | | | | | | |
| 15–24 | 79 | 4.28 | 4.23 | 4.03 | 4.16 | 4.17 | 4.11 | 4.10 | 4.11 | 3.94 | 4.19 | 4.06 | 4.22 | 4.09 | 4.15 |
| 25–34 | 98 | 4.14 | 4.32 | 4.03 | 4.19 | 4.17 | 4.12 | 4.26 | 4.19 | 3.92 | 4.18 | 4.05 | 4.19 | 4.09 | 4.14 |
| 35–44 | 85 | 4.33 | 4.38 | 4.21 | 4.31 | 4.31 | 4.19 | 4.25 | 4.22 | 3.96 | 4.40 | 4.18 | 4.29 | 4.25 | 4.27 |
| 45–54 | 94 | 4.43 | 4.46 | 4.17 | 4.12 | 4.29 | 4.10 | 4.19 | 4.14 | 4.03 | 4.48 | 4.26 | 4.27 | 4.18 | 4.22 |
| 55–64 | 36 | 4.58 | 4.47 | 4.39 | 4.28 | 4.43 | 4.22 | 4.42 | 4.32 | 4.08 | 4.50 | 4.29 | 4.58 | 4.44 | 4.51 |
| **Race** | | | | | | | | | | | | | | | |
| Malay | 268 | 4.35 | 4.45 | 4.16 | 4.31 | 4.32 | 4.21 | 4.30 | 4.26 | 4.04 | 4.40 | 4.22 | 4.31 | 4.25 | 4.28 |
| Chinese | 50 | 4.16 | 4.22 | 4.00 | 3.98 | 4.09 | 4.00 | 4.10 | 4.05 | 4.00 | 4.16 | 4.08 | 4.08 | 4.08 | 4.08 |
| Indian | 13 | 4.23 | 3.85 | 4.00 | 3.92 | 4.00 | 4.15 | 4.23 | 4.19 | 3.62 | 4.00 | 3.81 | 4.23 | 3.85 | 4.04 |
| Others | 61 | 4.33 | 4.20 | 4.15 | 3.95 | 4.16 | 3.92 | 3.97 | 3.94 | 3.74 | 4.26 | 4.00 | 4.28 | 4.00 | 4.14 |

Noted: red highlighted was moderate level.

Based on race group, 10 out of 10 of the mean variables of the importance of LH in high level (4: agree) in the Malay race responded. In the Chinese race, it was just a variable SC4 at a moderate level (3: partially agree). In the Indian race, four out of 10 of the mean was in moderate level (3: partially agree) was SC2, SC4, EN1, and PS2. Lastly, in other races, four out of 10 of the mean was in moderate level (3: partially agree) was SC4, EC1, EC2, and EN1. It is recommended in creating an effective education that every variable at a moderate level (3: partially agree) in the race must be highlighted and taken into consideration to increase the knowledge and awareness in CBE for LH.

Most of the mean of the importance of LH in high level (4: agree) just some contributions in moderate level (3: partially agree), especially in race group. The Indian race was at a moderate level (3: partially agree) in environmental contributions, meanwhile other races who were at a moderate level (3: partially agree) were in economic contributions. It must be highlighted or taken into consideration in creating an effective education to increase the knowledge and awareness in CBE for LH.

*4.2. Respondents' Attitude, Cultural Knowledge, and Awareness of the Participation Level*

There is six variables of the participation level, which is two variables in the low level (L1 and L2), two variables in the middle level (M1 and M2), and two variable in the high level (H1 and H2). Two out of six variables mean at moderate level (3: partially agree), meanwhile others were at low level (2: disagree).

The overall means of attitude, cultural knowledge, and awareness of the participation level are based on respondents' perspectives are represented in Table 8. The highest mean variable in high participation level (decision-making) was H1: I am interested in volunteering and participating from the beginning until the end, with a 3.18 mean (blue highlighted) in the moderate level (3: partially agree). Meanwhile, the lowest mean variable in the middle participation level (collaboration) was M2: I meet with local authorities and state government officials to discuss the issues, with a 2.51 mean (red highlighted) in the low level. Most respondents lack collaboration with local authorities and state government officials in discussing LH education and conservation issues. This makes information regarding LH education and conservation very low or late received in the local community.

**Table 8.** The overall means of attitude, cultural knowledge, and awareness of the participation level are based on respondents' perspectives (*n* = 392).

| Participation Level | Code | Variable | Mean | SD |
|---|---|---|---|---|
| **Low: L** (Information) | L1 | I get involved and keep up with the news regarding this conservation. | 3.01 | 1.11 |
| | L2 | I am familiar with this conservation. | 2.83 | 1.08 |
| **Middle: M** (Collaboration) | M1 | I receive information and do what local authorities and state government officials ask. | 2.98 | 1.17 |
| | M2 | I meet with local authorities and state government officials to discuss the issues. | 2.51 | 1.17 |
| **High: H** (Decision Making) | H1 | I am interested in volunteering and participating from the beginning until the end. | 3.18 | 1.20 |
| | H2 | I in my community have the power to change the decisions taken by local authorities and state government officials. | 2.77 | 1.31 |

The Comparison Means of Attitude, Cultural Knowledge, and Awareness in Gender, Age Level, and Race of the Participation Level on Low, Middle, and High Levels

Table 9 shows the comparison means of attitude, cultural knowledge, and awareness in gender, age level, and race of the participation level based on respondents' perspectives on low, middle, and high levels. Most of the mean of the participation level is in the low level (2: disagree), just some variables in the moderate level (3: partially agree). Participation was the most important part of doing a project to be successful, so the increase of participation level to ACTIVE in CBE for LH conservation must provide the best information and practical education to attract local community involvement and empowerment.

**Table 9.** The comparison means of attitude, cultural knowledge, and awareness in gender, age, and race of the participation level is based on respondents' perspectives on low, middle, and high levels.

| | N | L1 | L2 | L | M1 | M2 | M | H1 | H2 | H |
|---|---|---|---|---|---|---|---|---|---|---|
| **Overall** | 392 | 3.01 | 2.83 | 2.92 | 2.98 | 2.51 | 2.75 | 3.18 | 2.77 | 2.97 |
| **Gender** | | | | | | | | | | |
| Male | 193 | 2.98 | 2.83 | 2.91 | 2.96 | 2.53 | 2.75 | 3.21 | 2.80 | 3.00 |
| Female | 199 | 3.04 | 2.83 | 2.94 | 3.01 | 2.48 | 2.74 | 3.15 | 2.73 | 2.94 |
| **Age Level** | | | | | | | | | | |
| 15–24 | 79 | 3.23 | 2.99 | 3.11 | 3.13 | 2.63 | 2.88 | 3.63 | 3.11 | 3.37 |
| 25–34 | 98 | 2.99 | 2.87 | 2.93 | 3.12 | 2.62 | 2.87 | 3.24 | 2.88 | 3.06 |
| 35–44 | 85 | 3.14 | 2.86 | 3.00 | 3.12 | 2.68 | 2.90 | 3.34 | 2.78 | 3.06 |
| 45–54 | 94 | 2.79 | 2.63 | 2.71 | 2.68 | 2.21 | 2.45 | 2.72 | 2.44 | 2.58 |
| 55–64 | 36 | 2.89 | 2.89 | 2.89 | 2.78 | 2.28 | 2.53 | 2.78 | 2.53 | 2.65 |
| **Race** | | | | | | | | | | |
| Malay | 268 | 3.05 | 2.86 | 2.95 | 3.10 | 2.57 | 2.84 | 3.32 | 2.81 | 3.07 |
| Chinese | 50 | 3.04 | 2.88 | 2.96 | 2.84 | 2.44 | 2.64 | 3.28 | 3.00 | 3.14 |
| Indian | 13 | 3.00 | 2.92 | 2.96 | 3.15 | 2.54 | 2.85 | 3.15 | 2.85 | 3.00 |
| Others | 61 | 2.84 | 2.67 | 2.75 | 2.54 | 2.26 | 2.40 | 2.44 | 2.34 | 2.39 |

Noted: red highlighted was low level.

In the gender group, males only have one variable in the moderate level (3: partially agree) in high participation level H1 compared to females with three variables in the moderate level (3: partially agree) L1, M1, and H1. Its recommendation is to focus more on the male gender to improve the participation level in creating an effective education of CBE for LH.

Six out of six of the mean participation levels in the age level groups more senior in 45–55 and 55–64 have a low participation level (2: disagree) compared to a middle age level group of 15–24, 25–34, and 35–44. More interesting activities and interactive education need to focus on the age level groups 45–55 and 55–64 in creating an effective education of CBE for LH.

Based on the race group, six out of six of the mean variables of the participation level in the low level (2: disagree) in other races responded compared to the Malay, the Chinese, and the Indian races. Its recommendation in creating an effective education that every variable in participation level in the race must be highlighted, especially in other races to increase the knowledge and awareness in CBE for LH.

In creating an effective education, the L2 and M2 variable must be focused on every gender, age level, and race group of the participation level because it means in the low level (2: disagree). Meanwhile, the middle level (collaboration) of participation level must be focused on every gender, age level, and race group of the participation level because it means also in the low level (2: disagree).

## 5. Recommendation

**H1.** *There are significant participation levels and the importance of living heritage*.

Pearson product correlation of the participation level and the importance of LH was found to be very low positive in Table 10, and statistically significant (r = 0.254, $p < 0.001$). H1 was supported. This shows that an increase in the participation level to ACTIVE would lead to a higher altitude, cultural knowledge, and awareness of the importance of LHs in the local community.

**Table 10.** Correlation analysis of the participation level and the importance of LH.

|  | Participation Level | Importance of LH |
|---|---|---|
| **Participation level** | 1 | 0.254 ** |
| **Importance of LH** | 0.254 ** | 1 |

** Correlation is significant at the 0.01 level (2-tailed).

Table 11 summarizes the analysis of the level of participation and the importance of LH. To increase cultural knowledge and awareness in both categories, the male gender is the one that is given more attention. Several studies have found that, on average, males do better on general knowledge assessments than females [70–72], but this study argues against that finding since the differences are most likely the result of various interests that male and female each has.

**Table 11.** Summary analysis of the participation level and the importance of LH.

|  | The Importance of LH | Participation Level |
|---|---|---|
| Gender | Focused more on the male gender. | Focused more on the male gender. |
| Age Level | More focused on a middle-aged level group of 15–24, 25–34, and 35–44 to increase cultural knowledge and awareness. | Focused on the age level groups 45–55 and 55–64. Creating more interesting activities and interactive education. |
| Race | The Indian race needs to focus on environmental contributions meanwhile other races in economic contributions. | Every variable in participation level in the race must be highlighted, especially in other races. |
| Overall | The EN1 variable must be highlighted in every gender, age, and race group the importance of LH to increase cultural knowledge and awareness. | The L2 and M2 variable must be highlighted in every gender, age, and race group of the participation level, especially the middle level (collaboration). |

At the age level group, there are comparisons in the importance of LH and the participation level. This is the importance of LH the focused on a middle-age level group of 15–24, 25–34, and 35–44 to increase cultural knowledge and awareness compared to the participation level, focused on the age level groups 45–55 and 55–64 to create more interesting activities and interactive education. In addition, individuals in the oldest age

group reported role loss more frequently than participants in the younger age groups. Reduced physical capacity, the development of disease, and functional limitations, may all be linked to decreased participation level, which is more common among older age groups [73].

The Indian race needs to focus on environmental contributions meanwhile, other races in economic contributions in the importance of LH. Meanwhile, every variable in participation level in the race must be highlighted, especially in other races.

The importance of LH to increase cultural knowledge and awareness, while the L2: I am familiar with this conservation, and M2: I meet with local authorities and state government officials to discuss the issues, must be highlighted for overall analysis. The EN1: Knowledge and practice of cultural heritage accumulated over time to make sustainable use of natural resources and minimize the impact of climate change must be highlighted in every gender, age level, and race group (collaboration).

The new direction for future investigations is the CBE Framework for LH conservation and the community participation level toward sustainable development at the World Heritage Site (WHS) in Malaysia.

## 6. Conclusions and Implications

In conclusion, increasing participation in ACTIVE would lead to greater attitudes, cultural knowledge, and awareness of the importance of LHs in the community. The overall analysis of participation levels and the importance of LH as a general guide to raising community awareness, cultural understanding, and altitudes. If this LH is not maintained, the community suffers losses in terms of identity, image, sense of place, and sense of pride. Therefore, further research is required to determine the research topic of heritage educational programs, preferred learning styles, and teaching methods in CBE for LH conservation.

Due to resource constraints (time and financial), this research study only managed to research into one case study, the Melaka WHS, with returned and answered questionnaires by 392 local community. One of the most challenging and time-consuming parts of the field research exercise was due to the COVID-19 situation, the face-to-face data collection needed to change to online. The feedback from the local community took about half a year to complete. However, based on the amount of feedback and commitments received, it is more than sufficient to generalize the results, and therefore, the result highlighted in this research study is hopefully found to be trustworthy to represent the population of the Melaka WHS. An in-depth evaluation of cultural heritage education and current practices of heritage site management for the living heritage sites in Malaysia should be carried out by researchers. Indeed, this research study was based on one case study in one state in Malaysia only. In order to enhance research findings, a more thorough study needs to be carried out in every state in Malaysia where there are living heritage sites. This will prove whether the problems of cultural knowledge and participation level in cultural heritage education are similar or unique only to Melaka WHS, based on the findings from the other states. Although the recommendation aspect was highlighted in the results have shown that, it could be 'implementable', the details of the implementation aspects were not discussed because this recommendation would need to be studied in depth on the suitability and problems of implementation in real practice.

**Author Contributions:** Conceptualization, N.A.A.A. and N.F.M.A.; methodology, N.A.A.A., N.F.M.A. and N.A.I.; software, N.A.A.A.; validation, N.F.M.A. and N.A.I.; formal analysis, N.A.A.A.; investigation, N.F.M.A.; resources, N.A.A.A. and N.F.M.A.; data curation, N.A.A.A. and N.F.M.A.; writing—original draft preparation, N.A.A.A.; writing—review and editing, N.F.M.A. and N.A.A.A.; visualization, N.A.A.A.; supervision, N.F.M.A., N.A.I. and A.A.; project administration, N.F.M.A. and N.A.A.A.; funding acquisition, N.F.M.A., N.A.I. and A.A. All authors have read and agreed to the published version of the manuscript.

**Funding:** The Fundamental Research Grant Scheme (FRGS/1/2019/WAB04/UPM/02/2) of the Ministry of Higher Education of Malaysia provided funding for this study. The authors also acknowledge the support from Universiti Putra Malaysia.

**Institutional Review Board Statement:** Not applicable.

**Informed Consent Statement:** Not applicable.

**Data Availability Statement:** Data is contained within the article.

**Conflicts of Interest:** The authors declare no conflict of interest.

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
