# Peer review of "Community Participation in the Importance of Living Heritage Conservation and Its Relationships with the Community-Based Education Model towards Creating a Sustainable Community in Melaka UNESCO World Heritage Site"

_sustainability, doi:10.3390/su15031935_

Round 1

Reviewer 1 Report

Thank you very much for the opportunity to review this paper. I got an email from the editor that they got sufficient reviews from other peers. Thus, I would like to attach some comments that I made before receiving the email from the editor. Here, I want to stress that, this is only for the partial comments on your paper. Please go through the following comments to make a better-quality paper.

The research gap and the necessity of the research in the introduction have not been presented. 

Figures one and two are hard to read. Using too many figures makes less visibility of the paper.

Maslow's five needs have been mentioned but how it is linked with your article has not been discussed. (page 6).

2.2.3 is better to include in the method section.

Materials and methods

The data collection period has not been mentioned.

No discussions have been made on the data. Why these results are received should be discussed. What implication can be brought

The conclusion section is too short. No limitations are mentioned and what are the implications of this study and future research directions? Consistency and inconsistency have not been mentioned which is necessary for a paper.

References should thoroughly be checked and formatted. For example, number 30, 36, 10, 17, 18, 26 are not consistent.

Reviewer 2 Report

I would advise the author(s) to revise the text in line 38 and to replace "things" with "elements" or another more appropriate noun;

In line 47 "surroundings" is not the best option; try to replace it with another noun that better captures the idea of the environment as community/area of origin etc.;

Perhaps the authors can find a better term than "extraneous components";

The 2nd Title should be modified: Community-Based Education (CBE) Model: literature review;

Figures 1 and 2 need explanatory texts.

Does Figure 2 have a source?

Rephrase lines 211 and 212: Therefore, safeguarding intangible cultural heritage is essential. Sustains it, passes it on to succeeding generations, and gives it the flexibility to adapt to any situation, keeping it alive.

Is Maslow's pyramid indeed the best choice? Aren't there any other more suitable models?

I would suggest the author(s) to include Hofstede's cultural dimensions in their analysis - in section 2.2.2 (https://www.hofstede-insights.com/country-comparison/malaysia/)

The 3rd section (Materials and Methods) should only focus on the research methodology - explaining the research goals and hypotheses, the sampling method, the research tools, the analysis methods. Please move the cases at the beginning of the 4th section Results and Discussions.

Moreover, as long as the author(s) also present case studies, they must discuss in the 3rd section (Materials and Methods) the use of the case study method as well. Methodological choices ought to be explained.

In line 391 replace "samples" with "subjects/respondents".

In line 393 replace "chosen at random" with "chosen randomly".

Review the Methodological section and paragraphs 3.1 and 3.2, and move the information referring to the sampling method, sample description and research tool in the appropriate section (methodology). 

Table 3 contains rather old data. Try to update it.

Furthermore, the survey tool (3.3) and data analysis (3.4) should also be moved to the Materials and Methods section. It would be good to provide more information on the statistical analysis that you have undertaken (explain the tests that you have run).

I would advise the authors to rename section 4. Results and Discussions

How was the research conducted? Were surveys self-administered? Did you use operators?

Line 485: compared to females with that high level -> revise

I fail to properly catch the meaning of:  Therefore, more focus on the male gender must be 486 highlighted to provide information in CBE for LH to create an effective education (lines 486-487)

Please rephrase lines 488-489: "has a comparison in the age level group"

The entire paragraph from lines 488-493 needs to be clarified. Similarly, try o rewrite and clarify the next paragraph (lines 494-501 and 522-526). All sentences need predicates.

Regarding data in Table 7 and the results in lines 548-558, how many of the respondents are directly involved in tourism and hospitality related activities or have jobs related to direct decision taking? What about the involvement of the public local authorities and the policies developed at national level?

In lines 660-661: greater altitudes, cultural knowledge - is "altitudes" the right term?

It would be appropriate to extend the Conclusions sections. 

I believe that the references can be updated. If I am not mistaken, there are no sources newer than 2019. I guess the topic has been addressed more recently, too.

Author Response

Please refer to the document attachment for the revision.

Reviewer 3 Report

Dear authors, your paper is interesting. However, much work is still needed to improve the paper and make it publishable.

-The aims in the Abstract «It used the Community-Based Education Model in this study to gather 19 information on Living Heritage Conservation toward creating a Sustainability Community» must be explained in a different form (for instance using the word "aim", "goal" or "objective") in order that the reader knows that you talking about objectives. And objectives is not what one has done, but what one intends to achieve with the study.

-The explanation, kind of literature review of the topics of the study is ok.

- The reason you give for choosing the questions is not valid «The questions were chosen because it related to the category of people's opinions, values, attitudes, and behavior» or is not conveniently explained. How did you come to the questions of your survey. Which were the questions? Were they grounded on Literature review? This is critical.

Results presentation is OK.

The Discussion section is poor and unacceptable. It mixes elements of the Methods with Results. example of text that refers to methods «Reliability is the measure of the internal consistency of the constructs in this study. A construct is reliable if the Alpha (α) value is greater than .70 [65]. Construct reliability 611 was assessed using Cronbach’s Alpha.». Example of text that refers to Results «The results revealed that the participation level scale with six items (α = .882) and the importance of the LH scale with ten items (α = .944). Reliability results are shown in Table 9.».  The results section should make an interpretation of the results and compare them with other relevant studies in the topic and showing if they differ from previous studies, if they confirm previous studies, etc. This has not been done.

Please bear in mind that the last part of each study is where the contributions of the study can be evidenced. In the case of your paper, the final part is a disaster. You started the paper very well and as you approached the final part, it seems you lost momentum. The Conclusion section has only seven lines and one of the conclusions is that more research is needed. The so what question remains unanswered: What is this research important for?  What are ist contributions? Who can profit from this study?

I suggest that you create a Section titled Conclusions and Implications. There, you have the conclusions (the answers to the objectives of the research) and the explanation of the theoretical and the practical contributions of this paper, as well as its limitations and the suggestions to future research. 

Round 2

Reviewer 1 Report

The authors almost considered my comments. Thus, I am satisfied. I still suggest changing
Figure 1 because it is hard to read.

Author Response

Dear Reviewer 1,

I already changing the figure 1.

Thank you for your comment and suggestion.

Reviewer 2 Report

-

Author Response

Dear Reviewer 2,

Thank you for your comment and suggestions.

Reviewer 3 Report

Dear authors,

Thanks for accepting my suggestions. The paper has improved and I am satisfied with the results.

Best regards

Author Response

Dear Reviewer 3,

Thank you for your comment and suggestions.

Best Regard.